# A versatile platform for precise synthesis of asymmetric molecular brush in one shot

Binbin Xu [1], Chun Feng[1] & Xiaoyu Huang[1]

Asymmetric molecular brushes emerge as a unique class of nanostructured polymers, while their versatile synthesis keeps a challenge for chemists. Here we show the synthesis of well-defined asymmetric molecular double-brushes comprising two different side chains linked to the same repeat unit along the backbone by one-pot concurrent atom transfer radical polymerization (ATRP) and Cu-catalyzed azide/alkyne cycloaddition (CuAAC) reaction. The double-brushes are based on a poly(Br-acrylate-alkyne) homopolymer possessing an alkynyl for CuAAC reaction and a 2-bromopropionate initiating group for ATRP in each repeat unit. The versatility of this one-shot approach is demonstrated by CuAAC reaction of alkynyl/poly(ethylene oxide)-$N_3$ and ATRP of various monomers. We also show the quantitative conversion of pentafluorophenyl ester groups to amide groups in side chains, allowing for the further fabrication of diverse building blocks. This work provides a versatile platform for facile synthesis of Janus-type double-brushes with structural and functional control, in a minimum number of reactions.

---

[1] Key Laboratory of Synthetic and Self-Assembly Chemistry for Organic Functional Molecules, Shanghai Institute of Organic Chemistry, Chinese Academy of Sciences, 345 Lingling Road, Shanghai 200032, People's Republic of China. Correspondence and requests for materials should be addressed to X.H. (email: xyhuang@sioc.ac.cn)

As an important topic in polymer science, the synthesis of polymers with well-defined compositions and structures has attracted significant attention over the past several decades[1, 2]. Numerous complex polymers with well-organized architectures such as multiblock[3], star[4] and comblike[5] polymers have been developed, due to their extensive applications from engineering materials to medical devices[6, 7]. Among these complex polymers, molecular brushes represent a special type of polymers, in which multiple polymeric chains are densely attached to a linear polymer backbone, i.e., every repeat unit of the linear polymeric backbone possesses an attached polymeric side chain[8, 9]. Because of their compact branched structure and persistent cylindrical shape, such polymers have emerged as a unique class of nanostructured polymers[10–12]. Along with their structures, molecular brushes are attractive candidates for a variety of applications, including pH-sensitive probes[13], molecular pressure sensors[14] and smart surface coatings[15]. Recently, asymmetric molecular brushes, bearing different dense side chains attached to the polymeric backbone, have emerged[16–19]. Because of their special structures and strong intramolecular interaction of different side chains, such asymmetric molecular brushes exhibit very different properties in comparison with ordinary molecular brushes, and can be employed as nanostructured materials and giant surfactants[20–24].

Molecular brushes can be constructed via the grafting-from[25–27] and grafting-onto[28–30] approaches from a pre-formed polymeric backbone, or the grafting-through[31, 32] method using a macromonomer. The grafting-from strategy has been the most popular method for constructing well-defined molecular brushes, often with initiating groups for controlled radical polymerization on the pre-formed polymer. In contrast, the grafting-onto method is rarely employed since it often leads to a relatively low grafting density. Gao and Matyjaszewski[32] showed that this limitation could be overcome by the use of thin linear poly(ethylene oxide)-N$_3$ (PEO-N$_3$) chains ($M_n = 775$ g mol$^{-1}$) to couple with poly(2-hydroxyethyl methacrylate)-alkyne (PHEMA-alkyne) and obtained molecular brushes with a high grafting density (88%).

A special case of asymmetric molecular brushes are those in which two polymers of different composition are attached to each repeat unit along the main chain. One can think of these as Janus bottle brush polymers (asymmetric molecular double-brushes)[33–35]. The identical distribution of hetero-brushes along the backbone of asymmetric molecular double-brush could facilitate the self-assembly and allow access to diverse morphologies that are difficult to achieve for traditional asymmetric molecular brushes[36]. Moreover, with numerous amphiphilic junctions per macromolecule, asymmetric molecular double-brushes can be used as stabilizers of biphasic systems and hydrophobic solutes via noncovalent interactions, and Janus nanomaterials[24, 37]. However, there is very few robust method for efficient synthesis of such polymers due to their inherently challenging synthesis. Even so, some intelligent multi-step methods have been developed for the construction of asymmetric molecular double-brushes. Cheng et al. used the grafting-through approach to synthesize a series of polymers in which each repeat unit carried both a polystyrene (PS) and a polylactide (PLA) side chains. They used ring-opening metathesis polymerization (ROMP) to polymerize a PS-norbornene-PLA diblock macromonomer, which was first prepared in one-pot from a multifunctional agent bearing a hydroxyl as an initiating site for PLA and a trithiocarbonate functionality for reversible addition-fragmentation chain transfer (RAFT) polymerization of styrene[18]. Xie et al. also used a ROMP-based grafting-through approach to prepare an asymmetric polynorbornene brush with poly(2-(dimethylamino)ethyl methacrylate) (PDMAEMA) and poly(ε-caprolactone) (PCL) side chains based on a

heterotrifunctional inimer possessing two initiating sites (OH and Br) for ring-opening polymerization (ROP) and atom transfer radical polymerization (ATRP)[38]. Zhao et al. prepared asymmetric molecular brushes based on a poly(glycidyl methacrylate) backbone with PEO and poly(styrene-b-N-iso-propylacrylamide) (PS-b-PNIPAM) side chains[39]. The epoxy groups in the backbone were transformed to introduce azide groups for click coupling to install a RAFT agent and hydroxyls to which PEO side chains were grafted by esterification, and PS-b-PNIPAM side chains were then formed by RAFT polymerization.

One should note that two of these reports were limited to polynorbornene backbones obtained via ROMP in which the macromonomer required multiple steps for its synthesis. These included ROP of cyclic monomers to introduce PLA or PCL. One could in principle modify the norbornene monomer with polymers produced by RAFT or ATRP, but this would require even more synthetic steps. The third report described poly(glycidyl methacrylate) generated by ATRP, but then several synthetic steps were needed to introduce the PEO and PS-b-PNIPAM side chains. It remains a challenge to synthesize a variety of different asymmetric molecular brushes with pairs of different polymers attached to each repeat unit.

Here we report a highly efficient strategy for synthesizing Janus-type well-defined asymmetric molecular double-brushes. By this way, one can construct a variety of different Janus-type asymmetric molecular double-brushes (double-brush copolymers) using a minimum number of reactions, without a need for subsequent polymeric functionality transformations. These polymers can be prepared with controlled molecular weights and narrow molecular weight distributions.

## Results

**Synthetic route for asymmetric molecular double-brushes.** In designing our synthesis strategy, we envisioned a bifunctional homopolymer precursor with an alkynyl in each repeat unit for a grafting-onto reaction and an ATRP-initiating group in each repeat unit for grafting-from polymerizations. The Cu-catalyzed azide/alkyne cycloaddition (CuAAC) click reaction is well-known as a highly efficient coupling reaction with a high tolerance of functionality and insensitivity to various solvents[40]. ATRP is a versatile polymerization technique that allows the preparation of molecular brushes with controlled architecture, predictable degree of polymerization and narrow molecular weight distribution[41]. Since that both ATRP and CuAAC reaction employ the very similar catalytic systems, we hypothesize that the combination of ATRP and CuAAC click reaction could be utilized to directly synthesize asymmetric molecular brushes in a one-shot system[42, 43], that is to say, the copper(I)/ligand could mediate two different reactions (ATRP and CuAAC coupling) simultaneously for a concerted synthesis of asymmetric molecular brushes.

In this work, a very special homopolymer precursor, poly(Br-acrylate-alkyne), is constructed in the first step (Supplementary Fig. 1). The repeat unit in backbone has a 2-bromopropionate initiating group for ATRP as well as an alkynyl for click coupling. We show that in the second step, two independent polymeric chains can be attached to each repeat unit by simultaneous (one-shot) Cu-catalyzed ATRP and CuAAC click coupling reaction (Fig. 1). This is a polymeric analog of a bifunctional small molecule agent, in which a block copolymer consisting of two different polymeric chains can be generated in a single pot reaction.

**Synthesis of Br-acrylate-alkyne trifunctional monomer.** The trifunctional acrylate monomer, which we refer to

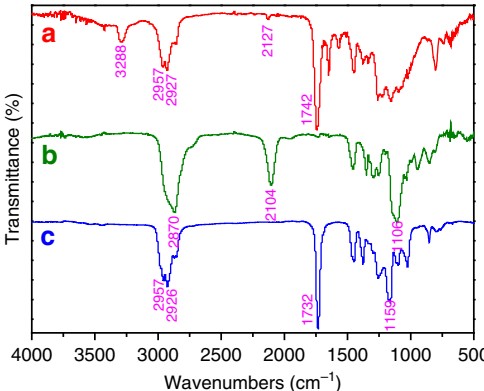

**Fig. 1** Construction of asymmetric molecular double-brushes. Synthetic route of PA-*g*-PEA/PEO, PA-*g*-PDMAEA/PEO, PA-*g*-PS/PEO and PA-*g*-PPFMA/PEO asymmetric molecular double-brushes in one-shot system via the combination of ATRP and CuAAC click reaction

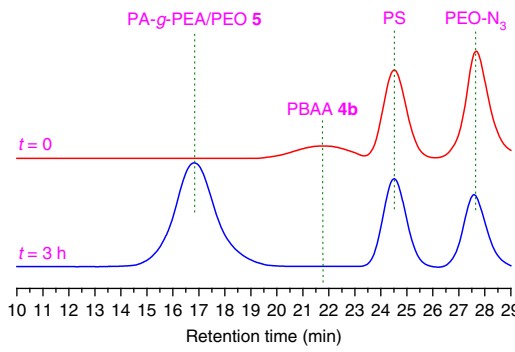

**Fig. 2** Chemical structure of polymers. Fourier transform infrared (FT-IR) spectra of PBAA **4** (**a**), PEO-N$_3$ (**b**) and PA-*g*-PEA/PEO **5** asymmetric molecular double-brush (**c**).

**Fig. 3** Monitoring of CuAAC click reaction. GPC traces of ATRP/CuAAC system for PA-*g*-PEA/PEO **5** asymmetric molecular double-brush before (*red*) and after (*blue*) the reaction, feeding ratio: [-N$_3$]: [-C≡CH] = 4:1 and [EA]:[-Br] = 150:1, internal standard: PS ($M_n$ = 3790 g mol$^{-1}$)

Br-acrylate-alkyne (BAA), was synthesized in four steps (Supplementary Fig. 1). In the first step, we used a Baylis-Hillman reaction to introduce a hydroxymethyl into *tert*-butyl acrylate (*t*BA) to obtain *tert*-butyl (2-hydroxymethyl)acrylate. Next, the hydroxymethyl was converted to a 2-bromopropyl ester via carbodiimide coupling to yield the *tert*-butyl 2-((2-bromo-propanoyloxy)methyl)acrylate (*t*BBPMA) **1** monomer[44]. Then the *tert*-butoxycarbonyl was hydrolyzed to the carboxyl using CF$_3$COOH in dichloromethane at ambient temperature, followed by esterification reaction with propargyl bromide in DMF to afford the target Br-acrylate-alkyne monomer **3**. The chemical structure of monomer **3** was characterized by $^1$H (Supplementary Fig. 2A) and $^{13}$C (Supplementary Fig. 3) NMR. The resonance signals of the double bond appeared at 6.00 and 6.48 p.p.m. ($^1$H NMR), and 129.0 and 134.1 p.p.m. ($^{13}$C NMR), respectively. The peaks located at 4.42 and 2.50 p.p.m. belonged to 1 proton of C*H*Br and 1 proton of alkynyl, respectively, which demonstrated the introduction of corresponding functionalities for ATRP and CuAAC click reaction. Thus, these points clearly demonstrate the successful synthesis of Br-acrylate-alkyne monomer.

**Preparation of poly(Br-acrylate-alkyne) bifunctional macro-agent.** The polymer with bifunctional repeat units [poly(Br-acrylate-alkyne) (PBAA) **4**] was synthesized by RAFT polymerization of the monomer **3** in toluene at 70 °C using 2,2'-azobis(isobutyronitrile) (AIBN) as the initiator and cumyl

dithiobenzoate (CDB) as the chain transfer agent (CTA). We intentionally limited the extent of monomer conversion to avoid any possible chain coupling or crosslinking, with a final polymer yield on the order of 60%. Then the dithiobenzoate end group was removed by AIBN at 65 °C[45]. Two well-defined samples of PBAA **4** were obtained (Supplementary Table 1) with narrow molecular weight distributions (**4a**: $M_n$ = 3100 g mol$^{-1}$, $M_w/M_n$ = 1.28, degree of polymerization (DP): 11; **4b**: $M_n$ = 7900 g mol$^{-1}$, $M_w/M_n$ = 1.30, DP: 21). Both gel permeation chromatography (GPC) curves showed a unimodal and symmetrical elution peak (Supplementary Fig. 4), indicative of right choice of CDB as CTA to mediate RAFT polymerization of monomer **3**.

Polymer **4** was characterized by $^1$H NMR (Supplementary Fig. 2B) and Fourier transform infrared (Fig. 2a). The proton resonance signal of the double bond (5.00–7.00 p.p.m.) disappeared after polymerization, and the signal of two protons of -C*H$_2$*C- in the polyacrylate backbone appeared at 2.09 p.p.m., confirming the successful polymerization of monomer **3**. The peak d at 4.25 p.p.m. in the $^1$H NMR spectrum of polymer **4** is attributed to one proton of O$_2$COC*H*(CH$_3$)Br moiety and the resonance signal of one proton of alkynyl is located at 2.62 p.p.m. (peak c). These signals show that both the ATRP-initiating sites and click reaction sites were preserved during the RAFT polymerization. The RAFT polymerization mechanism is also verified by the minor peaks between 7.15 and 7.36 p.p.m. corresponding to five protons of the terminal phenyl originating from the CTA of CDB. Using this signal, the absolute degree

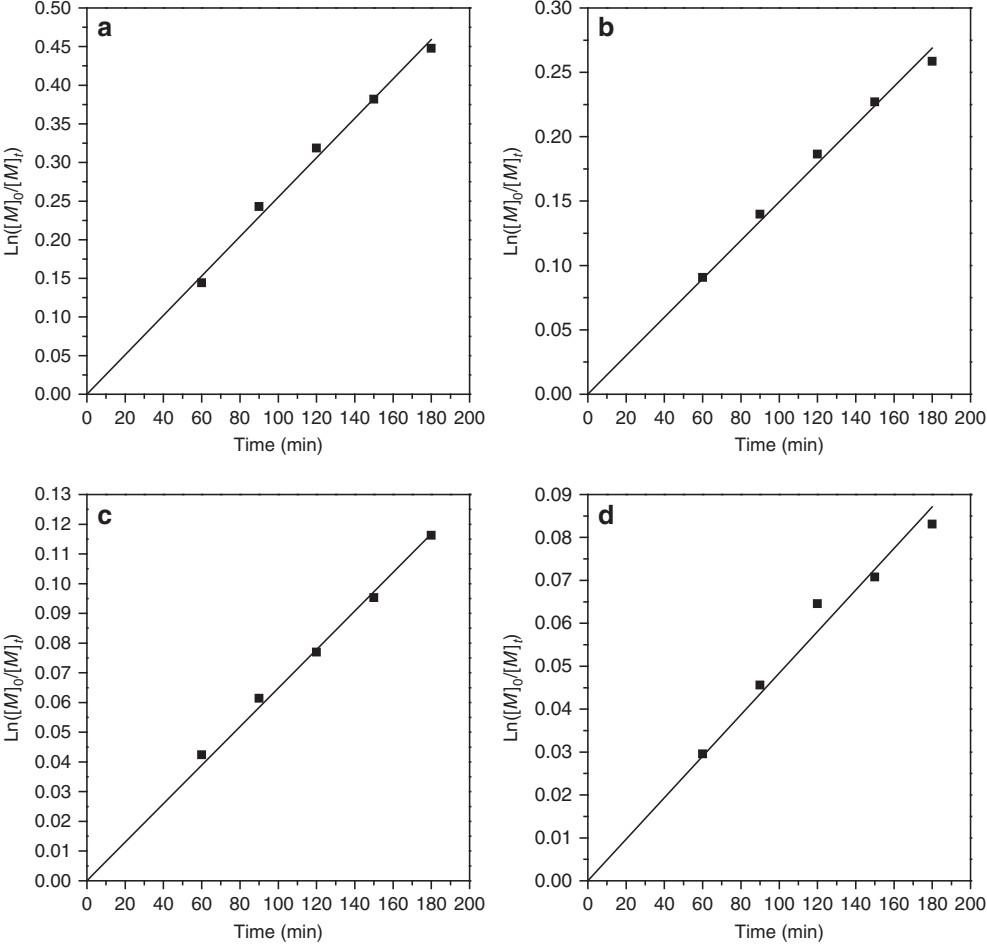

**Fig. 4** ATRP kinetics. Kinetic plot for ATRP of EA **a**, DMAEA **b**, St **c** and PFMA **d** initiated by PBAA **4b**

of polymerization of monomer **3** and the molecular weight ($M_{n,NMR}$) can be estimated, as summarized in Supplementary Table 1. Further structural information of polymer **4** is available from the FT-IR spectrum. The signals at 3288 and 2127 cm$^{-1}$ in FT-IR spectrum confirm the existence of the alkynyl in polymer **4**. Taken together, these data support the formation of well-defined bifunctional macro-agent **4**, which consists of an ATRP-initiating site and a CuAAC click reaction site in each repeat unit.

**One-shot synthesis of asymmetric molecular double-brush.** In our polymer design, we envision Cu-catalyzed click coupling of a polymer to the pendant alkynyls in the repeat units simultaneous with ATRP polymerization from the 2-bromopropionate groups. To begin, we first constructed a polymeric azide. To simplify our experiments and to focus on the versatility of the ATRP polymerization, we began with the same PEO-N$_3$[32] as that used by Matyjaszewski et al.[32] to graft PEO onto a polymer backbone derived from PHEMA, and then extended our efforts to a longer PEO-N$_3$. The azide-terminated PEO-N$_3$ samples were obtained with a two-step protocol, starting with commercial PEO-OH ($M_n = 750$ and 2000 g mol$^{-1}$), and the presence of terminal azide was verified by the typical FT-IR signal at 2104 cm$^{-1}$ (Fig. 2b).

The first one-shot dual reaction experiment used PBAA **4b** as the macroinitiator and ethyl acrylate (EA) as the monomer in the ATRP polymerization. The ATRP of EA and the CuAAC click reaction of PEO-N$_3$ with PBAA **4b** were carried out at 60 °C in DMF, with the feeding ratios of [-N$_3$]:[-C≡CH] of 4:1 and [EA]:

[-Br] of 150:1 (see Supplementary Table 2). One should recall that in the macroinitiator [-C≡CH] = [-Br].

In this reaction, we wanted not only to monitor the increase in molecular weight of the asymmetric molecular double-brush but also to monitor the extent of conversion of PEO-N$_3$, which was present in excess. We could follow both processes by GPC through the use of a polystyrene internal standard (PS, $M_n = 3790$ g mol$^{-1}$). Since the PS internal standard would not participate in either reaction, it served as a reference for the elution peak of PEO-N$_3$ ($M_n = 775$ g mol$^{-1}$). One can see in Fig. 3 that the four peaks of PEO-N$_3$, the PS standard, the starting polymer **4b**, and the asymmetric molecular double-brush do not overlap. Therefore, the percentage of PEO-N$_3$ reacted could be calculated by comparing the normalized peak areas of PEO-N$_3$ and PS before and after the reaction. In this example, ~23% of PEO-N$_3$ reacted after 3 h (Supplementary Table 2), and thus we calculated that the extent of grafting of PEO side chains is 92% (Supplementary Table 2), similar to the value reported by Matyjaszewski et al.[32] (88%). We learn that almost all of the pendant alkynyls of PBAA **4b** were coupled to PEO side chains via the CuAAC click reaction. In support of this conclusion, we found that the FT-IR signals of the alkynyl had disappeared (Fig. 2c).

To gain more understanding of the kinetics of the CuAAC click reaction, we also repeated the reactions under similar conditions but for shorter times. After 1 h we found 52% conversion of the pendant alkynyls, and after 2 h we found 80% conversion (Supplementary Table 2). When we used PEO-N$_3$ with longer PEO chain ($M_n = 2000$ g mol$^{-1}$), we found 68% conversion of the pendant alkynyls after 3 h and 76% conversion after 5 h. By

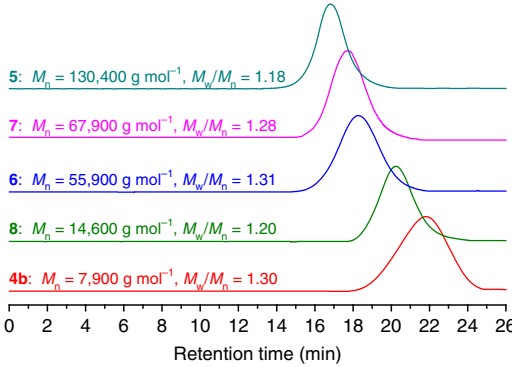

**Fig. 5** Molecular weight of precursor and double-brushes. GPC traces of PBAA **4b** macro-agent, PA-*g*-PEA/PEO **5**, PA-*g*-PDMAEA/PEO **6**, PA-*g*-PS/PEO **7** and PA-*g*-PPFMA/PEO **8** asymmetric molecular double-brushes in tetrahydrofuran (THF)

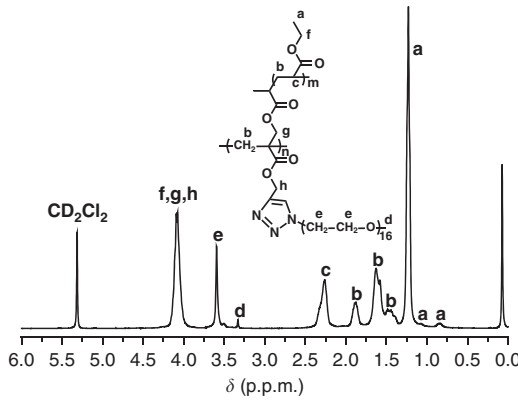

**Fig. 6** Chemical structure of asymmetric molecular double-brush. $^1$H NMR spectrum of PA-*g*-PEA/PEO **5** asymmetric molecular double-brush in CD$_2$Cl$_2$

varying the reaction time, we can control the extent of PEO grafting.

We also examined the kinetics of the ATRP polymerization of EA. We used gas chromatography (GC) to monitor monomer conversion (see experiment details in Supplementary Methods). We found a linear plot of $\ln([M]_0/[M]_t)$ vs. time (Fig. 4a). This first-order polymerization kinetics is a typical characteristic of ATRP[46] and indicates a constant propagating radical concentration during the polymerization. This result indicates that the ATRP of EA can also be controlled in the current one-shot system. The combination of these observations indicates that the preparation of polyacrylate-*g*-poly(ethyl acrylate)/poly(ethylene oxide) (PA-*g*-PEA/PEO) **5** asymmetric molecular double-brush can be carried out with excellent control of the PEA side chain length. Control is in principle possible by varying either the reaction time or the initial feeding ratio of EA monomer to macro-agent **4**. It also should be noted that because of the chain end fidelity of ATRP[46], the PEA side chains can be extended with other monomers suitable for ATRP to yield block side chains.

We next repeated this chemistry using methyl methacrylate (MMA) as the monomer. On one hand, we wanted to establish the generality of the ATRP reaction for polymer **4**. In addition, we used this reaction to learn about the initiation efficiency of the ATRP reaction. The initiation efficiency of ATRP is normally evaluated by NMR. In the case of the polymerization of EA, the $^1$H NMR signal of the CH$_2$C*H*Br end group of PEA side chain (~4.30 p.p.m.) overlapped with other signals so that it was difficult to determine the initiation efficiency. With MMA as the monomer, we examined the asymmetric molecular double-brush by $^{13}$C NMR. Here we found complete disappearance of the CH$_3$*C*HBr signal at 40.1 p.p.m. that was present in the PBAA **4b** macro-agent. In its place, we could detect the CH$_2$*C*(CH$_3$)Br signal for the group at the end of PMMA chain. Thus, within the limits of signal integration in the $^{13}$C NMR measurements, we infer complete conversion of the ATRP-initiating groups in the MMA reaction. By analogy, we deduce nearly complete CH$_2$C*H*Br initiation for the ATRP of EA in the preceding example. Further evidence for this conclusion is provided by the absence of a shoulder or significant tailing in the GPC curve of PA-*g*-PEA/PEO **5**.

The purified product, which we refer to as PA-*g*-PEA/PEO **5**, was characterized by GPC and $^1$H NMR. The GPC curve of PA-*g*-PEA/PEO **5** shows a unimodal and symmetrical elution peak (dark cyan line in Fig. 5) with a higher molecular weight ($M_n = 130,400$ g mol$^{-1}$) compared to that of PBAA **4b** ($M_n = 7900$ g mol$^{-1}$) and a narrow molecular weight distribution ($M_w/M_n = 1.18$), indicative of the negligible side reactions such as

intermolecular coupling reactions[47]. In the present work, we used a high feeding ratio of EA to the ATRP-initiating group (150:1) to suppress possible intermolecular coupling, and we used a relatively high feeding ratio of azide functionality to alkyne (4:1) to improve the efficiency of CuAAC coupling reaction.

Figure 6 shows the $^1$H NMR spectrum of PA-*g*-PEA/PEO **5** and all proton resonance signals of both PEO and PEA side chains appear in the spectrum. The signals at 3.31 (peak d) and 3.58 (peak e) p.p.m. originate from the OC*H*$_3$ end group and OC*H*$_2$C*H*$_2$ repeat unit in PEO side chains, respectively. The signals at 0.85, 1.08, 1.21 (peak a) and 2.24 (peak c) p.p.m. correspond to three protons of CO$_2$CH$_2$C*H*$_3$ and two protons of CH$_2$C*H* in PEA side chains, respectively. Therefore, the mean number of EA repeat units ($m$) in PEA side chain can be estimated from the integration areas of peak d ($S_d$) and c ($S_c$). From Eq. 1 (where 92% represents the grafting extent of PEO side chains), we calculate a mean degree of polymerization of 46 EA units per PEA chain. Thus the absolute molecular weight ($M_{n,NMR}$) can be calculated according to Eq. 2 (where 6000 is the $M_{n,NMR}$ value of PBAA **4b** (Supplementary Table 1), 21 is the DP of monomer **3** in PBAA **4b** (Supplementary Table 1), 100 and 775 are the molecular weights of EA monomer and PEO-N$_3$). The value is 117,800 g mol$^{-1}$, much different from that obtained from GPC (130,400 g mol$^{-1}$), which may be attributed to the complex branch structure[48].

$$m = (3 \times 92\% \times S_c)/S_d \qquad (1)$$

$$M_{n,NMR} = 6,000 + 21 \times m \times 100 + 21 \times 92\% \times 775 \qquad (2)$$

These results verify the successful one-shot synthesis of a well-defined PA-*g*-PEA/PEO **5** asymmetric molecular double-brush possessing dense PEO and PEA side chains linked to each repeat unit along the polymeric backbone, with a nearly complete ATRP initiation efficiency and a very high grafting efficiency of PEO side chains.

**Construction of asymmetric molecular double-brushes with other monomers**. Compared to previous synthetic protocols[38, 39] for asymmetric molecular brushes, the current system starting from PBAA **4** is simple and versatile because well-defined asymmetric molecular double-brushes can be obtained in only one step without polymeric functionality transformations, just using four reagents (macro-agent **4**, ATRP monomer, PEO-N$_3$, and catalyst: copper(I)/ligand).

**Table 1 Synthesis of asymmetric molecular double-brushes[a]**

| Entry | Monomer | Temperature (°C) | Time (h) | $M_{n,GPC}$[f] (g mol$^{-1}$) | $M_w/M_n$[f] | $m$[g] | $M_{n,NMR}$[h] (g mol$^{-1}$) |
|---|---|---|---|---|---|---|---|
| 5[b] | EA | 60 | 3.0 | 130,400 | 1.18 | 46 | 117,800 |
| 6[c] | DMAEA | 40 | 4.0 | 55,900 | 1.31 | 24 | 91,000 |
| 7[d] | St | 80 | 4.0 | 67,900 | 1.28 | 29 | 85,700 |
| 8[e] | PFMA | 80 | 5.0 | 14,600 | 1.20 | 10 | 75,200 |

[a]PBAA **4b**: $M_{n,GPC}$ = 7900 g mol$^{-1}$, $M_w/M_n$ = 1.30; PEO-N$_3$: $M_n$ = 775 g mol$^{-1}$, solvent: DMF
[b]Feeding ratio: [EA]:[-N$_3$]:[-Br]:[CuBr]:[PMDETA] = 150:4:1:2:2, [-C≡CH] = [-Br], reaction time: 3 h
[c]Feeding ratio: [DMAEA]:[-N$_3$]: [-Br]:[CuBr]:[HMTETA] = 200:4:1:2:2, [-C≡CH] = [-Br], reaction time: 4 h
[d]Feeding ratio: [St]:[-N$_3$]:[-Br]:[CuBr]:[HMTETA] = 200:4:1:2:2, [-C≡CH] = [-Br], reaction time: 4 h
[e]Feeding ratio: [PFMA]:[-N$_3$]:[-Br]:[CuBr]:[PMDETA] = 150:4:1:2:2, [-C≡CH] = [-Br], reaction time: 5 h
[f]Measured by GPC at 35 °C in THF. [g]The number of repeated unit per side chain (PEA, PDMAEA, PS and PPFMA) obtained from $^1$H NMR. [h]Obtained from $^1$H NMR

We tested the versatility of this platform to prepare diverse asymmetric molecular double-brushes by examining the ATRP of various monomers (2-(dimethylamino)ethyl acrylate (DMAEA), styrene (St) and pentafluorophenyl methacrylate (PFMA)) under similar conditions (Table 1). Here DMAEA is a N-containing hydrophilic acrylate monomer, styrene is a widely used vinyl monomer, and PFMA is an efficient building block for post-polymerization functionalization. Using a constant initial feeding ratio of PEO-N$_3$ to alkynyl (4:1), the grafting extent of PEO side chain of different asymmetric molecular double-brushes was significantly influenced by the temperature. We used a reaction temperature of 40 °C for the preparation of polyacrylate-g-poly((2-(dimethylamino)ether acrylate)/poly(ethylene oxide) (PA-g-PDMAEA/PEO) **6**, and 80 °C for the preparation of polyacrylate-g-polystyrene/poly(ethylene oxide) (PA-g-PS/PEO) **7** and polyacrylate-g-poly(pentafluorophenyl methacrylate)/poly (ethylene oxide) (PA-g-PPFMA/PEO) **8**. Because of the high efficiency of CuAAC click reaction at 60 °C, we only monitored the conversion of PEO-N$_3$ for the preparation of PA-g-PDMAEA/PEO **6** at a lower temperature of this reaction. Even at 40 °C, the CuAAC click reaction maintained a high efficiency: the conversion of PEO-N$_3$ was 20% within 4 h and the grafting extent of PEO side chain of PA-g-PDMAEA/PEO **6** was as high as 80%, still comparable with that of simple molecular brushes[32].

Moreover, PA-g-PDMAEA/PEO **6**, PA-g-PS/PEO **7** and PA-g-PPFMA/PEO **8** all show unimodal and symmetrical elution peaks in the GPC curves (Fig. 5), with low polydispersities (≤ 1.31) and much higher molecular weights compared to that of macro-agent **4b**. PA-g-PDMAEA/PEO **6**, PA-g-PS/PEO **7** and PA-g-PPFMA/PEO **8** were characterized by $^1$H NMR, $^{13}$C NMR and FT-IR (see Supplementary Information). Supplementary Figs. 5–7 present the $^1$H NMR spectra of PA-g-PDMAEA/PEO **6**, PA-g-PS/PEO **7** and PA-g-PPFMA/PEO **8**, respectively, displaying all proton resonance signals of both EO and ATRP monomer (DMAEA, St or PFMA) repeat units. The conversion of the ATRP monomer at appropriate time intervals was also measured by GC, which revealed that the ATRP of DMAEA, St and PFMA in the one-shot system all obeyed a first-order rate law (Fig. 4b–d). In short, this one-shot synthesis protocol shows the robust nature in direct synthesis of asymmetric molecular brushes via ATRP of various monomers.

**Post-polymerization modification of PA-g-PPFMA/PEO.** Despite the robust nature of ATRP toward the direct polymerization of various monomers, the current one-shot system is obviously not suitable for monomers potentially incompatible with ATRP process or monomers that require special ATRP conditions[49]. Here, post-polymerization modification can be used effectively to obtain these functional polymers[50]. PPFMA[51, 52] has been widely used as an efficient building block for polymer post functionalization. It reacts quantitatively with amines under mild conditions and

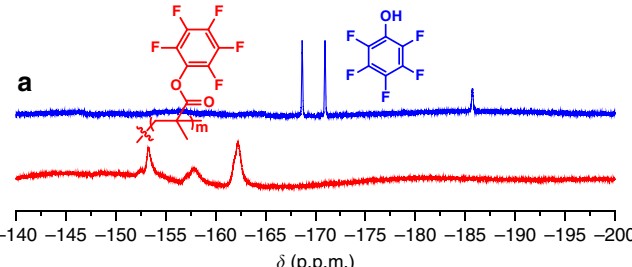

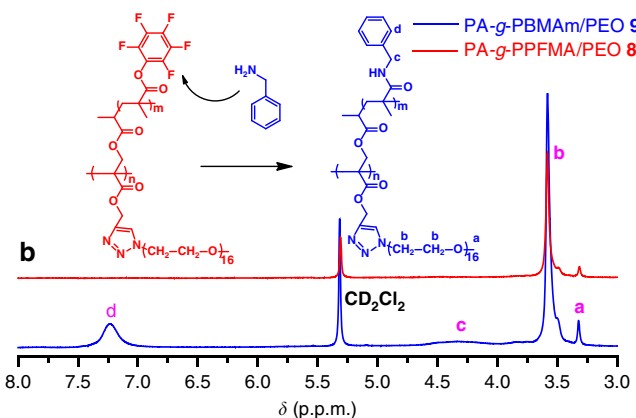

PA-g-PBMAm/PEO **9**
PA-g-PPFMA/PEO **8**

**Fig. 7** Monitoring of the aminolysis reaction. $^{19}$F (**a**) and $^1$H (**b**) NMR spectra of PA-g-PPFMA/PEO **8**/benzylamine system before (red) and after (blue) the reaction in CD$_2$Cl$_2$

the reaction can be monitored by $^{19}$F NMR. Encouraged by contributions from the Theato group[51, 52], we hypothesized that this chemistry could be employed as a supplementary platform for indirect synthesis of more asymmetric molecular double-brushes by using PFMA monomer for ATRP.

In the current case, we examined the aminolysis of PA-g-PPFMA/PEO **8** with benzylamine at 50 °C in tetrahydrofuran (THF) (Supplementary Fig. 8), and monitored the reaction by $^{19}$F NMR. It can be seen in Fig. 7a that the typical resonance signals corresponding to PPFMA side chains disappeared after the aminolysis (blue line), while only the signals attributed to pentafluorophenol appeared after the reaction, confirming the complete removal of pentafluorophenyl ester groups. The introduction of benzyl was demonstrated by $^1$H NMR, in which a new signal originating from five protons of -C$_6$H$_5$CH$_2$ in the N-benzyl methacrylamide (BMAm) repeat unit appeared at 7.25 p.p.m. (peak d of blue line) after the aminolysis (Fig. 7b). The $^1$H NMR spectrum after the aminolysis (blue line in Fig. 7b) also shows that the signals of PEO side chains were preserved during the aminolysis reaction.

The product of aminolysis, polyacrylate-g-poly(N-benzyl methacrylamide)/poly(ethylene oxide) (PA-g-PBMAm/PEO) **9**, shows a unimodal and symmetrical elution peak by GPC (Supplementary Fig. 9) with a similar molecular weight and a similar molecular weight distribution compared to that of PA-g-PPFMA/PEO **8**. This result also affirms that the polymeric skeleton was retained during the aminolysis. It is worthy noting that the product of the modification has poly(meth)acrylamide side chains, not poly(meth)acrylate side chains, demonstrating that aminolysis of PPFMA side chains can be an alternative for introducing diverse poly(meth)acrylamide side chains into the brush copolymer. This post-polymerization modification strategy based on PA-g-PPFMA/PEO enables a facile and efficient approach under mild conditions for affording a broader range of asymmetric molecular brushes not limited to poly(meth)acrylate side chains, without affecting PEO side chains or the backbone.

## Discussion

We have presented a facile synthesis of well-defined asymmetric molecular double-brushes with controlled architecture and narrow molecular weight distributions ($M_w/M_n \leq 1.31$). On the basis of the preparation of PBAA, a bifunctional macro-agent consisting of two different active sites (alkynyl and Br), a series of asymmetric molecular double-brushes was synthesized in a one-shot system by concurrent grafting-from via ATRP and a highly efficient grafting-onto via CuAAC click coupling. The ATRP allowed precise size control in the side chains via the reaction time and feeding ratio, whereas the extent of CuAAC click grafting could be controlled via the reaction time. The combination of the versatility of ATRP and the high efficiency of CuAAC click reaction can meet the demand for the synthesis of complex polymeric materials and significantly broaden the library of accessible polymer structures, either asymmetric or symmetric molecular double-brush polymers. More importantly, we demonstrate the possibility of post-polymerization modification of asymmetric molecular double-brushes possessing pentafluorophenyl ester groups in the side chains. This approach allows for further precise fabrication of a broad variety of polymer building blocks in which one avoids the direct ATRP of monomers that are difficult to polymerize or require special polymerization conditions.

In summary, this work not only reveals the excellent compatibility between copper-catalyzed ATRP and CuAAC click reactions, but provides a versatile platform for the facile synthesis of complex asymmetric molecular brushes with precise structure and functionality control, in a minimum number of reaction steps and reactants, without polymeric functionality transformations.

**Data availability**. The authors declare that the data supporting the findings of current work are available within the paper and its Supplementary Information files, and also are available from the authors upon reasonable request.

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

## Acknowledgements

We thank the financial supports from National Basic Research Program of China (2015CB931900), National Key Research and Development Program of China (2016YFA0202900), National Natural Science Foundation of China (21474127 and 21632009), Strategic Priority Research Program of Chinese Academy of Sciences (XDB20020000), Youth Innovation Promotion Association of Chinese Academy of Sciences (2016233) and Shanghai Scientific and Technological Innovation Project (14JC1493400, 16JC1402500, 14520720100 and 16520710300). We thank Prof. Mitchell A. Winnik (University of Toronto) for his constructive advice and assistance in polishing the manuscript.

## Author contributions

X.H. conceived and designed the experiments. B.X. and C.F. performed the experiments. B.X. and C.F. analyzed the data. C.F. contributed reagents/materials/analysis tools. B.X. and X.H. wrote the paper.
