## [Peer Review File · Nature Communications]

Reviewers' Comments:

Reviewer #1:

Remarks to the Author:

The work by Huang et al. is dedicated to the area of molecular brush to obtain the well-defined asymmetric molecular dual-brushes by one-shot synthetic protocol. Although the molecular brush systems have been already known largely, the asymmetric molecular dual-brushes synthesized by two separated or tandem polymerization methods were rarely reported before. What is more important is that the investigation on the control over the polymerization or reaction behavior of different monomers or reagents with the precise regulation of the brush length and density in one-shot way seems to be original, possessing the necessary novelty and diversity. Data and figures are of good quality. All new compounds and polymers are characterized with a set of necessary methods. Taking into account everything mentioned above, the presented results are scientifically reliable and reproducible. I will therefore recommend this manuscript for publication after the following minor points being addressed.

(1) The used PEO length is actually short, is this one-shot strategy suitable for longer PEO chain?

(2) For ATRP initiated by the macro-agent with multifunctional active center, the crosslinking reaction is affected by the conversion of monomers. How much conversion of monomers is permitted to avoid the crosslinking reaction?

Reviewer #2:

Remarks to the Author:

This manuscript reports the synthesis of brush copolymers with two different types of grafts by simultaneous ATRP and click reaction. Overall, this is a solid research work in polymer field. The research design is quite fine, and experimental work was well done. However, the novelty of this manuscript is not as high as that general readers would expect for articles published on Nature Communications. This work is considered as incremental work based on previous studies in the field, instead of an important breakthrough in polymer research, for the following reasons: 1) the synthesis of brush copolymers with two different types of grafts has been reported by multiple groups using various approaches (including one example of one-shot synthesis from backbone precursor); 2) compatibility of ATRP and click reaction allowing for simultaneous ATRP-click process for polymer synthesis has also been reported by multiple groups; 3) the preparations of grafts of brush polymers by either ATRP "grafting from" or click "grafting onto" have been well documented. Thus, although this is a solid work with an innovative research design based on a nice combination of recent advances in polymer chemistry for the synthesis of a special class of molecular brush, it is more suitable for publishing on a specialized journal, instead of Nature Communications. Other minor comments are as follows:

1) The manuscript highlighted "one-shot" process for the asymmetrical molecular brushes; however, this "one-shot" synthesis only occurred at grafting stage, and polymeric precursors were needed. Thus, without indicating starting chemicals properly, the use of "one-shot" is confusing and sometimes misleading. For example, in the first sentence of abstract, "The first one-shot synthesis of well-defined asymmetric molecular dual-brushes with well-tuned side chain length and density, using a Br-acrylate-alkyne trifunctional monomer possessing an alkynyl for CuAAC "click" reaction and a 2-bromopropionate initiating group for ATRP, is reported", readers may easily understand that Br-acrylate-alkyne trifunctional monomer was a starting chemical used in the "one-shot" synthesis. But it is incorrect because actually the trifunctional monomer needs to be converted to the corresponding polymer at first before grafting (i.e. two sequential steps from the trifunctional monomer to the molecular brush).

2) The abbreviations, such as ATRP, RAFT and St, need to be defined at first before using.

3) The abbreviations of the brush copolymers, such as b-PEA-PEO, are inappropriate and did not

follow any nomenclature. The composition of backbone polymer also needs to be given in an abbreviation. Following the general nomenclature for grafted copolymers, such brush copolymers may be abbreviated as P1-g-P2/P3, where P1 is the backbone polymer and P2 and P3 are polymer grafts. Poly(Br-acrylate-alkyne), the backbone polymer, may be abbreviated as PBAA.

4) Page 8, the DP values of the two Poly(Br-acrylate-alkyne) macro-agents should be given along with Mw/Mn values.

5) Page 13, the claim of 100% initiation efficiency by ATRP grafting from process is incorrect. The disappearance of the resonance signal of CH₃CHBr at 40.1 ppm in poly(Br-acrylate-alkyne) 4b macro-agent only indicated the full consumption of the initial CH₃CHBr moieties (side reactions, such as biradical coupling or disproportionation, may occur along with initiation). If one really wishes to probe grafting density (essentially the same as initiation efficiency), the grafts formed by ATRP need to be detached and carefully analyzed.

6) Significant figures of experimental data need to be considered carefully. Experimental data derived by NMR analysis should only keep two significant figures. GPC data should only keep three significant figures.

Reviewer #3:

Remarks to the Author:

The authors reported the synthesis of a new trifunctional monomer and the subsequent preparation of dual brushes. This is a very elegant combination of ATRP and CuAAC for the precise introduction of two different side chains into the same repeating unit of the backbone in a very simple but common one-shot system. The authors clearly show the 100% initiation efficiency and the very high (>90%) coupling efficiency. I think this work is a significant contribution to polymer chemistry because this platform will ignite the versatile synthesis of dual brushes, which can be applied in many fields such as chemistry and materials science. The article is well-organized and well-written. Therefore, I recommend it for publication in Nat Commun if the following questions are addressed.

Questions:

1. For the new monomer 3, FT-IR and MS data are needed.
2. The name of b-PEA-PEO is confusing. Technically, this is a graft copolymer, not a block polymer. Please clarify it.
3. Scheme 1: In this one-pot system, different reactions (monomers) need different ligands, not just PMDETA. CuBr/ligand instead of PMDETA/CuBr is more suitable.
4. Figure 4 caption: "of" in the third line is unnecessary.
5. Figure 6: The name of dual brush 9 (b-PBMA-PEO) should be consistent with the name (b-PBMAm-PEO) in the article.

Point-by-point answer to reviews

For Reviewer 1:

The work by Huang et al. is dedicated to the area of molecular brush to obtain the well-defined asymmetric molecular dual-brushes by one-shot synthetic protocol. Although the molecular brush systems have been already known largely, the asymmetric molecular dual-brushes synthesized by two separated or tandem polymerization methods were rarely reported before. What is more important is that the investigation on the control over the polymerization or reaction behavior of different monomers or reagents with the precise regulation of the brush length and density in one-shot way seems to be original, possessing the necessary novelty and diversity. Data and figures are of good quality. All new compounds and polymers are characterized with a set of necessary methods. Taking into account everything mentioned above, the presented results are scientifically reliable and reproducible. I will therefore recommend this manuscript for publication after the following minor points being addressed.

We thank the reviewer for these positive comments, particularly that our paper “possessing the necessary novelty and diversity” for publication in *Nature Communications*.

1. *The used PEO length is actually short, is this one-shot strategy suitable for longer PEO chain?*

Thanks for your kind comment. This strategy is also applicable for longer PEG chain, though the steric hindrance effect would become more pronounced with the increasing in the length of polymeric chain. We now include results with a longer PEO-N₃ ($M_n = 2,025$ g/mol) for the same reaction conditions employed previously. Here we found 68% conversion after 3 h and 76% conversion after 5 h. We are actually impressed by how efficient this reaction is. These data have been added to page 12 of the revised manuscript according to your nice advice.

2. *For ATRP initiated by the macro-agent with multifunctional active center, the*

crosslinking reaction is affected by the conversion of monomers. How much conversion of monomers is permitted to avoid the crosslinking reaction?

Thank you for your kind comment. According to previous reports (*Polymer* **2003**, *44*, 1449. *Macromol. Chem. Phys.* **2006**, *207*, 57. *Macromolecules* **2008**, *41*, 325. *Macromolecules* **2001**, *34*, 4375. *Macromolecules* **2003**, *36*, 1843.), both a high feeding ratio of the monomer to the ATRP initiating group (>150:1) and a low conversion of the monomer (<30%) are necessary to suppress the intermolecular crosslinking reaction in the synthesis of graft polymer. In the current work as listed in Table 1 (page 18), we employed high feeding ratios of the monomer to the ATRP initiating group (200:1 & 150:1) and kept low conversions of monomer (<30%, generally less than 15%) to avoid or minimize any crosslinking reactions.

For Reviewer 2:

This manuscript reports the synthesis of brush copolymers with two different types of grafts by simultaneous ATRP and click reaction. Overall, this is a solid research work in polymer field. The research design is quite fine, and experimental work was well done. However, the novelty of this manuscript is not as high as that general readers would expect for articles published on Nature Communications. This work is considered as incremental work based on previous studies in the field, instead of an important breakthrough in polymer research, for the following reasons: 1) the synthesis of brush copolymers with two different types of grafts has been reported by multiple groups using various approaches (including one example of one-shot synthesis from backbone precursor); 2) compatibility of ATRP and click reaction allowing for simultaneous ATRP-click process for polymer synthesis has also been reported by multiple groups; 3) the preparations of grafts of brush polymers by either ATRP "grafting from" or click "grafting onto" have been well documented. Thus, although this is a solid work with an innovative research design based on a nice combination of recent advances in polymer chemistry for the synthesis of a special class of molecular brush, it is more suitable for publishing on a specialized journal, instead of Nature Communications.

Thanks for your kind comments. We first appreciate the positive comments of “*this is a solid research work in polymer field, the research design is quite fine, and experimental work was well done*” and “*with an innovative research design based on a nice combination of recent advances in polymer chemistry for the synthesis of a special class of molecular brush*” very much.

In re-reading our original submission, we may not have been sufficiently clear in describing the novelty or originality of our contribution. In our work, we are able to prepare a single homopolymer precursor with two orthogonally reactive groups in each repeat unit. In a single step, we are able to add one polymer in a CuAAC grafting-onto reaction and a second polymer in an ATRP grafting-from reaction to generate two different side chains at each repeat unit of the backbone. This type of dual brush polymer has been reported previously, but by complex or cumbersome multistep syntheses. For example, multiple-step polymerization and post-polymerization functionalization approaches are reported (*Macromolecules* **2017**, *50*, 2201; *Macromolecules* **2010**, *43*, 7434; *Biomacromolecules* **2009**, *10*, 2033; *Macromolecules* **2007**, *40*, 9503.). Other approaches yield dual brush polymers with the different polymeric side chains distributed randomly along the backbone (*J. Am. Chem. Soc.* **2009**, *131*, 18525. *Macromolecules* **2006**, *39*, 7513. *Macromolecules* **2006**, *39*, 584. *Macromolecules* **2002**, *35*, 985.) or a block-style (*Macromolecules* **2011**, *44*, 9635. *J. Am. Chem. Soc.* **2013**, *135*, 4203. *ACS Macro Lett.* **2013**, *2*, 809.). The number of papers that we can cite here is an indication of the interest in this type of brush polymer. Our approach is simpler and more versatile, and very different from any of the examples cited above.

Moreover, given the great compatibility of ATRP with monomer structure and high efficiency of CuAAC “click” reaction with excellent functional compatibility, the novel strategy (general platform) for the facile synthesis of complex asymmetric polymer brush described in the current manuscript may give rise to a large degree of attention for the researchers in synthetic chemistry and materials science.

For reasons 1: “*the synthesis of brush copolymers with two different types of grafts*”

has been reported by multiple groups using various approaches (including one example of one-shot synthesis from backbone precursor)”.

Although a variety of polymer brushes have been reported, the reports on asymmetric dual brushes are rather rare, especially well-defined asymmetric dual brushes with two different side chains linked to each repeat unit of backbone. In order to prepare well-defined asymmetric dual brushes with two different side chain linked to each repeat unit of backbone, multiple-step strategy or post-polymerization functionalization are usually needed (*Macromolecules* **2017**, *50*, 2201. *Macromolecules* **2010**, *43*, 7434. *Biomacromolecules* **2009**, *10*, 2033. *Macromolecules* **2007**, *40*, 9503.). Those features will inevitably make the synthesis tedious and inefficient.

The “one-shot” synthesis referred to by the review is the work of Cheng *et al.* (ref 18) who reported the synthesis of asymmetric polymer brushes by ring-opening metathesis polymerization (ROMP) of a norbornene-based macromonomer to which polystyrene and polylactide side chains were attached. While we appreciate the importance of their contribution, it is very different and less versatile than the approach we describe in our submission.

Although acrylate-based monomer is kind of more popular and can be polymerized by widely used controlled radical polymerization, the preparation of asymmetric polymer brushes containing polyacrylate backbone is still painstaking and more challenging (*Macromolecules* **2011**, *44*, 9635. *Macromolecules* **2012**, *45*, 4623. *Macromolecules* **2008**, *41*, 9004.). As reported in a previous literature (*Macromolecules* **2008**, *41*, 9004.), “*This is crucial to gain access to the expected structure and is fortunately convenient for norbornene derivative monomer bearing two different functional groups on the norbornene ring, but it is difficult for vinyl monomers to construct the complex polymer architectures unless introducing a spacer between the vinyl end and the moiety with two functional groups, or yielding macroinitiator under strict reaction conditions.*”. Obviously, it is difficult for the preparation of vinyl monomers with different active sites to construct the complex polymer architectures without protective group chemistry and polymeric functionality

transformation.

Satisfyingly, in the present work, our starting materials are very simple, the polyacrylate-based bifunctional macro-agent consisting of two different active sites (alkynyl and Br) in each repeat unit can be readily synthesized in few synthetic steps. To our best knowledge, this bifunctional macro-agent is the only polyacrylate-based macro-agent for the one-shot preparation of asymmetric polymer brush. With the successful preparation of bifunctional macro-agent, different asymmetric polymer dual-brushes can be synthesized in a one-shot system by concurrent controlled ATRP with precise size control in the side chains via the time and feeding ratio, and highly efficient CuAAC “click” chemistry with precise grafting extent control via the time. The combination of versatility of ATRP and high efficiency of CuAAC “click” reaction can meet the demand for asymmetric polymer dual brushes with dense side chains. In addition, we demonstrate the possibility of post-polymerization modification of asymmetric polymer dual-brush with pentafluorophenyl ester groups in the side chains, which allows for a further precise fabrication of a broad variety of building block, avoiding the direct ATRP of tough monomers under complicated conditions or several tough monomers potentially incompatible with ATRP process.

Thus, our work presented in this paper introduces a novel synthetic way to prepare various known and unknown asymmetric polymer brushes by the combination of versatility of ATRP and high efficiency of CuAAC “click” reaction, which may provide readers a wide thinking on the possibilities for constructing a variety of complex polymeric materials and significantly broaden the library of polymer structures.

For reasons 2: *“compatibility of ATRP and click reaction allowing for simultaneous ATRP-click process for polymer synthesis has also been reported by multiple groups.”*

As the reviewer mentioned that the compatibility of ATRP and click reaction allowing for simultaneous ATRP-click process for polymer synthesis has been reported, we also noticed this point and some corresponding references are cited (ref. 37 and 38). The most attractive point in our current work we believe is the one-shot

synthetic approach for the preparation of asymmetric polymer brushes with two different side chains linked to each repeat unit of the backbone by the combination of ATRP and click reaction, not the compatibility of ATRP and click reaction itself. We also demonstrate the possibility of precise structural control of asymmetric polymer dual-brush through ATRP with precise size control in the side chains via the time and feeding ratio, and “click” chemistry with precise grafting extent control via the time.

For reasons 3: “*the preparations of grafts of brush polymers by either ATRP "grafting from" or click "grafting onto" have been well documented.*”

Although a variety of polymer brushes have been reported by either ATRP with “grafting-from” or click reaction with “grafting-onto”, the reports on asymmetric dual brushes are rather rare, especially well-defined asymmetric dual brushes with two different side chain linked to each repeat unit of the backbone. In order to prepare well-defined asymmetric dual brushes with two different side chain linked to each repeat unit of the backbone, multiple-step strategy or post-polymerization functionalization are usually needed. These features will inevitably make the synthesis tedious and inefficient. Although Cheng *et al.* reported the synthesis of asymmetric polymer brushes with each graft site carrying polystyrene (PS) and polylactide (PLA) side chains through the grafting-through approach via ROMP using a PS-NB-PLA diblock macromonomer, which was prepared from a multifunctional agent bearing a hydroxyl as an initiating site for ROP and a trithiocarbonate functionality for RAFT polymerization, by one-pot RAFT polymerization of styrene and ROP of lactide (ref. 18). However, this one pot method is different with our strategy, in which the macromonomer suitable for ROMP was employed.

Although acrylate-based monomer is kind of more popular and can be polymerized by widely used controlled radical polymerization, the preparation of asymmetric polymer brushes containing polyacrylate backbone is still painstaking and more challenging (*Macromolecules* **2011**, *44*, 9635. *Macromolecules* **2012**, *45*, 4623. *Macromolecules* **2008**, *41*, 9004.). As reported in a previous literature (*Macromolecules* **2008**, *41*, 9004.), “*This is crucial to gain access to the expected*

structure and is fortunately convenient for norbornene derivative monomer bearing two different functional groups on the norbornene ring, but it is difficult for vinyl monomers to construct the complex polymer architectures unless introducing a spacer between the vinyl end and the moiety with two functional groups, or yielding macroinitiator under strict reaction conditions.”. Obviously, it is difficult for the preparation of vinyl monomers with different active sites to construct the complex polymer architectures without protective group chemistry and polymeric functionality transformation.

In the present work, one of our major contributions is the homopolymer precursor that serves as the framework for the simultaneous “grafting-from” and “grafting-onto” reactions since that the polyacrylate-based bifunctional macro-agent consists of two different active sites (alkynyl and Br) in each repeat unit. To our best knowledge, this bifunctional macro-agent is the only polyacrylate-based macro-agent for the one-shot preparation of asymmetric polymer brush. We think this is really a big step forward. With the successful preparation of bifunctional macro-agent, different asymmetric polymer dual-brushes can be synthesized in a one-shot system by concurrent controlled ATRP with precise size control in the side chains via the time and feeding ratio, and highly efficient CuAAC “click” chemistry with precise grafting extent control via the time. The combination of versatility of ATRP and high efficiency of CuAAC “click” reaction can meet the demand for asymmetric polymer dual brushes with dense side chains. Furthermore, the polymerization of the pentafluorophenyl methacrylate monomer and the subsequent transformation of the PFPMA arms are particularly valuable.

1. *The manuscript highlighted "one-shot" process for the asymmetrical molecular brushes; however, this "one-shot" synthesis only occurred at grafting stage, and polymeric precursors were needed. Thus, without indicating starting chemicals properly, the use of "one-shot" is confusing and sometimes misleading. For example, in the first sentence of abstract, "The first one-shot synthesis of well-defined asymmetric molecular dual-brushes with well-tuned side chain length and density,*

using a Br-acrylate-alkyne trifunctional monomer possessing an alkynyl for CuAAC "click" reaction and a 2-bromopropionate initiating group for ATRP, is reported", readers may easily understand that Br-acrylate-alkyne trifunctional monomer was a starting chemical used in the "one-shot" synthesis. But it is incorrect because actually the trifunctional monomer needs to be converted to the corresponding polymer at first before grafting (i.e. two sequential steps from the trifunctional monomer to the molecular brush).

Thanks for your kind comments. We have extensively revised the manuscript including the abstract and we clarify what we mean by "one-shot" according to your nice advice.

2. The abbreviations, such as ATRP, RAFT and St, need to be defined at first before using.

Thanks for your helpful advice. We have defined all abbreviations at the first time they are used in the article according to your kind suggestion.

3. The abbreviations of the brush copolymers, such as b-PEA-PEO, are inappropriate and did not follow any nomenclature. The composition of backbone polymer also needs to be given in an abbreviation. Following the general nomenclature for grafted copolymers, such brush copolymers may be abbreviated as P1-g-P2/P3, where P1 is the backbone polymer and P2 and P3 are polymer grafts. Poly(Br-acrylate-alkyne), the backbone polymer, may be abbreviated as PBAA.

Thanks for your helpful advice. We have carefully renamed all asymmetric polymer dual-brushes according to your kind suggestion.

4. Page 8, the DP values of the two Poly(Br-acrylate-alkyne) macro-agents should be given along with Mw/Mn values.

We have added the corresponding information to page 8 according to your nice advice.

5. Page 13, the claim of 100% initiation efficiency by ATRP grafting from process is incorrect. The disappearance of the resonance signal of CH_3CHBr at 40.1 ppm in poly(Br-acrylate-alkyne) 4b macro-agent only indicated the full consumption of the initial CH_3CHBr moieties (side reactions, such as biradical coupling or disproportionation, may occur along with initiation). If one really wishes to probe grafting density (essentially the same as initiation efficiency), the grafts formed by ATRP need to be detached and carefully analyzed.

Thanks for your kind comments. In the current case (Table 1), a high feeding ratio of the monomer to the ATRP initiating group (200:1 & 150:1) and a low conversion of the monomer (<30%, generally less than 15%) are employed to avoid the crosslinking reaction. Moreover, there is no significant tailing or shoulders in GPC curves ($M_w/M_n \leq 1.31$), indicating negligible occurrence of side reactions (*Macromolecules* **2008**, *41*, 325. *Macromolecules* **2001**, *34*, 6883.). Given the similar structure of our macro-agent with previous reports and living/controlled characteristic of ATRP (*Macromolecules* **1998**, *31*, 9413. *Macromolecules* **2008**, *41*, 325. *Macromolecules* **2001**, *34*, 6883.), the initiation efficiency was assumed to be nearly 100% in the current case. In order to be precise, we have modified the corresponding description throughout the manuscript according to your nice advice.

6. Significant figures of experimental data need to be considered carefully. Experimental data derived by NMR analysis should only keep two significant figures. GPC data should only keep three significant figures.

Thanks for your helpful advice. We have checked previous literatures and carefully modified the significant figures of experimental data throughout the manuscript according to your kind suggestion.

For Reviewer 3:

The authors reported the synthesis of a new trifunctional monomer and the subsequent preparation of dual brushes. This is a very elegant combination of ATRP and CuAAC for the precise introduction of two different side chains into the same

repeating unit of the backbone in a very simple but common one-shot system. The authors clearly shown the 100% initiation efficiency and the very high (>90%) coupling efficiency. I think this work is a significant contribution to polymer chemistry because this platform will ignite the versatile synthesis of dual brushes, which can be applied in many fields such as chemistry and materials science. The article is well-organized and well-written. Therefore, I recommend it for publication in Nat Commun if the following questions are addressed.

We thank the reviewer for these positive comments to our paper, especially for the comment that “this work is a significant contribution to polymer chemistry because this platform will ignite the versatile synthesis of dual brushes”.

1. *For the new monomer 3, FT-IR and MS data are needed.*

The FT-IR and HR-MS data have been added in supporting information (page 5 & 6) according to your nice advice.

2. *The name of b-PEA-PEO is confusing. Technically, this is a graft copolymer, not a block polymer. Please clarify it.*

Thanks for your kind comment. In our current work, *b*-PEA-PEO is the asymmetric polymer dual-brush comprising two different side chains linked to each repeat unit of the backbone. It is not easy to give this kind of dual-brush polymers a proper name. We use this manner: *b*-side chains 1-side chains 2, that is, *b*-PEA-PEO means *brush*-poly(ethyl acrylate)-poly(ethylene oxide).

However, reviewer 2 also found our notation to be unclear, but suggested a new notation that clarifies the structure of our brushes. We have adopted this new notation as polyacrylate-*g*-poly(ethyl acrylate)/poly(ethylene oxide) (PA-*g*-PEA/PEO). Thus, we have revised all names of dual-brushes throughout the manuscript.

3. *Scheme 1: In this one-pot system, different reactions (monomers) need different ligands, not just PMDETA. CuBr/ligand instead of PMDETA/CuBr is more suitable.*

We have revised Scheme 1 according to your nice advice.

4. *Figure 4 caption: “of” in the third line is unnecessary.*

We have made this correction according to your kind suggestion.

5. *Figure 6: The name of dual brush 9 (b-PBMA-PEO) should be consistent with the name (b-PBMAM-PEO) in the article.*

We have made the corresponding revision according to your nice advice.

With best regards,

Sincerely yours,

Xiaoyu Huang

Reviewers' Comments:

Reviewer #1:

Remarks to the Author:

After reading the revised manuscript and the responsive letter carefully, I found that the authors respond the concerned issues from reviewers 1 and 2 one by one satisfactorily, and I therefore give my proposal to accept this work for publication. Of course, it should be better to give all the author names in each reference.

Reviewer #2:

Remarks to the Author:

This work on asymmetrical brush copolymers is solid with nice combination of known chemistries and design principles to develop a relatively versatile synthetic route. Because it does not reveal a new synthetic principle or report an unprecedented type of materials, this work is considered incremental with moderate overall novelty and significance. Thus specialized journals can be more suitable avenues to publish this work. Some improvements are needed for the current version of manuscript.

1) The term "dual-brushes" was used to refer the specific type of asymmetrical brush copolymers (see 1st sentence of abstract and other places). However, in literature "dual-brushes" typically refers to block-type brush copolymer (e.g. ACS Macro Lett., 2013, 2, 809–813; Polym. Chem., 2011, 2, 137). According to literature, such asymmetrical brush copolymers with a graft site carrying two heterografts may be better termed as "double-brushes" or "double-brush copolymers".

2) In the third paragraph of introduction, it should be explained more carefully why study of such type of asymmetrical brush copolymers is important. A better summary regarding the research work in this domain should be made regarding both synthetic methods and polymer properties.

3) The following important references on such asymmetrical brush copolymers are missed and should be cited:

- a) Macromolecules, 2017, 50, 2201.
- b) J. Am. Chem. Soc., 2016, 138, 11501.
- c) Polym. Chem., 2016, 7, 4476.
- d) J. Polym. Sci., Part A: Polym. Chem. 2014, 52, 3250.
- e) Macromolecules 2010, 43, 3153

Reviewer #3:

Remarks to the Author:

The quality of the revised manuscript has been significantly improved and all the questions have been satisfactorily addressed. Therefore, I recommend it for publication in this journal.

Point-by-point answer to reviews for NCOMMS-17-04852A

For Reviewer 1:

After reading the revised manuscript and the responsive letter carefully, I found that the authors respond the concerned issues from reviewers 1 and 2 one by one satisfactorily, and I therefore give my proposal to accept this work be publication. Of course, it should be better to give all the author names in each reference.

We thank the reviewer for these positive comments and we also will revise the references according to the style of *Nature Communications*.

For Reviewer 2:

1. *The term “dual-brushes” was used to refer the specific type of asymmetrical brush copolymers (see 1st sentence of abstract and other places). However, in literature “dual-brushes” typically refers to block-type brush copolymer (e.g. ACS Macro Lett., 2013, 2, 809–813; Polym. Chem., 2011, 2, 137). According to literature, such asymmetrical brush copolymers with a graft site carrying two heterografts may be bettered termed as “double-brushes” or “double-brush copolymers”.*

We have carefully renamed all asymmetric molecular dual-brushes as asymmetric molecular double-brushes throughout the manuscript according to your kind suggestion.

2. *In the third paragraph of introduction, it should be explained more carefully why study of such type of asymmetrical brush copolymers is important. A better summary regarding the research work in this domain should be made regarding both synthetic methods and polymer properties.*

We have added the following information “The identical distribution of hetero-brushes along the backbone of asymmetric molecular double-brush could facilitate the self-assembly and allow access to diverse morphologies that are difficult to achieve for traditional asymmetric molecular brushes.³⁶ Moreover, with numerous amphiphilic junctions per macromolecule, asymmetric molecular double-brushes can be used as stabilizers of biphasic systems and hydrophobic solutes via noncovalent interactions, and Janus nanomaterials.^{24,37} However, there is very few robust method for efficient synthesis of such polymers due to their inherently challenging synthesis.” to page 4 according to your nice advice.

3. *The following important references on such asymmetrical brush copolymers are missed and should be cited: a) Macromolecules, 2017, 50, 2201. b) J. Am. Chem. Soc., 2016, 138, 11501. c) Polym. Chem., 2016, 7, 4476. d) J. Polym. Sci., Part A: Polym. Chem. 2014, 52, 3250. e) Macromolecules 2010, 43, 3153*

We have inserted these literatures (ref. 33-37) according to your kind suggestion.

For Reviewer 3:

The quality of the revised manuscript has been significantly improved and all the questions have been satisfactorily addressed. Therefore, I recommend it for publication in this journal.

We thank the reviewer for these positive comments.

With best regards,

Sincerely yours,

Xiaoyu Huang